# A Narrative Review of Factors Associated with Skin Carotenoid Levels

**DOI:** 10.3390/nu15092156

**Published:** 2023-04-30

**Authors:** Matthew P. Madore, Jeong-Eun Hwang, Jin-Young Park, Seoeun Ahn, Hyojee Joung, Ock K. Chun

**Affiliations:** 1Department of Nutritional Sciences, University of Connecticut, Storrs, CT 06269, USA; matthew.p.madore@uconn.edu; 2Device Research Center, Samsung Advanced Institute of Technology (SAIT), Samsung Electronics Co., Ltd., Suwon 16678, Republic of Korea; je.hwang@samsung.com (J.-E.H.); jya.park@samsung.com (J.-Y.P.); 3Department of Public Health Science, Graduate School of Public Health, Seoul National University, Seoul 08826, Republic of Korea; hjjoung@snu.ac.kr; 4Institute of Health and Environment, Seoul National University, Seoul 08826, Republic of Korea; ase0821@snu.ac.kr

**Keywords:** fruits, vegetables, carotenoids, blood carotenoids, skin carotenoid, resonance Raman spectroscopy (RRS), reflective spectroscopy (RS), dietary guidelines

## Abstract

Despite consistent evidence that greater consumption of fruits and vegetables (FV) is associated with significant reductions in chronic disease morbidity and mortality, the majority of adults in the United States consume less than the amounts recommended by public health agencies. As such, there is a critical need to design and implement effective programs and policies to facilitate increases in FV consumption for the prevention of these diseases. To accomplish this, an accurate, inexpensive, and convenient method for estimating the dietary FV intake is required. A promising method for quantifying the FV intake via proxy that has gained interest in recent years is the measurement of skin carotenoid levels via spectroscopy-based devices. However, there exist certain dietary and non-dietary factors that may affect the skin carotenoid levels independently of the dietary intake of carotenoids. In order to validate the ability of this method to accurately estimate the FV intake among diverse demographics, these factors must be identified and taken into consideration. Therefore, this narrative review seeks to summarize the available research on factors that may affect the skin carotenoid levels, determine current gaps in knowledge, and provide guidance for future research efforts seeking to validate spectroscopy-measured skin carotenoid levels as a means of accurately estimating the FV intake among various populations.

## 1. Introduction

Perhaps, one of the most well-studied and established aspects of a health-promoting diet is a high intake of fruits and vegetables (FV). Numerous meta-analyses of observational studies and randomized controlled trials (RCTs) have found that greater FV intake is associated with lower risks of morbidity and mortality from non-communicable diseases, including obesity, various cancers, hypertension, diabetes, and cardiovascular and neurodegenerative diseases [1,2,3,4,5,6,7,8,9,10,11,12]. Expectedly, there has been a long-standing interest in identifying and characterizing the components of FV responsible for these beneficial effects, which has led to the discovery of an array of bioactive phytochemicals. At present, evidence from observational research and clinical trials suggests that alongside their low calorie density and fiber content, certain antioxidant and anti-inflammatory phytochemicals and their metabolites likely play appreciable roles in mediating the positive impacts of FV on non-communicable disease outcomes [13]. In particular, carotenoids, yellow, orange, and red pigments with antioxidant and anti-inflammatory properties found in a wide variety of FV, have a considerable body of literature supporting their beneficial effects on human health [14]. Consistent with evidence concerning FV intake, multiple meta-analyses of prospective cohort studies have found that greater intake of dietary carotenoids is associated with lower risks of cardiovascular disease, cancer, all-cause mortality, diabetes, and neurodegenerative diseases [15,16,17]. However, while there are an appreciable number of health benefits that may result from greater FV and carotenoid consumption, most adolescents and adults in the United States (US) do not meet the minimum recommended intakes [18,19]. Furthermore, it has been estimated that over 100,000 annual cardiometabolic deaths may be attributable to a suboptimal FV intake [20]. Accordingly, identifying groups with low FV intakes and developing strategies that promote their consumption is vital for chronic disease prevention.

To effectively identify populations with a low FV intake and to assess the efficacy of various approaches for promoting FV consumption, an accurate, inexpensive, and convenient method for estimating dietary FV intake is critical. At present, a few methods are commonly utilized, but they are not without important limitations. Survey-based methods such as 24 h dietary recalls and food frequency questionnaires are very convenient; however, they rely on memory and knowledge of serving sizes and are prone to a number of biases such as social desirability and recall bias [21,22]. Given their presence in a wide variety of (but not all) FV and the lack of endogenous synthesis in humans, carotenoid concentrations in the blood have also been used as an objective marker of dietary FV intake. While blood carotenoids are currently considered the best biological marker available, the need to collect and analyze blood samples using high-performance liquid chromatography (HPLC) or liquid chromatography–mass spectrometry (LC-MS) makes the process time-consuming and invasive and can result in selection bias, since many individuals may not want to have blood drawn [23,24].

Since the skin is also a reservoir for dietary carotenoids and the assessment of an individual’s skin carotenoid concentration only requires a simple scan using a spectroscopy device, it has garnered interest as another potential biomarker for the objective estimation of FV intake [25]. In recent years, studies have confirmed that skin carotenoid levels assessed via resonance Raman (RRS) and reflective spectroscopy (RS) devices correlate well with both self-reported FV intake and blood carotenoid concentrations, indicating that this approach may indeed offer a similarly reliable and less invasive way to ascertain carotenoid intake [26]. In brief, RRS quantifies skin carotenoids by assessing the Raman response originating from the vibrating carbon backbone common to carotenoids following excitation by a low-power laser, whereas RS quantifies skin carotenoids by measuring their absorption via reflection under application of topical pressure [27]. While the results of existing validation studies for these methods are promising, there are a number of potential factors that may affect the metabolism and tissue distribution of carotenoids and therefore require consideration if skin carotenoid levels are to be used to effectively estimate FV intake [28]. Therefore, this review aims to summarize the literature concerning factors that may affect skin carotenoid levels, identify existing uncertainties, and provide direction for future research aimed at further validating RRS- and RS-measured skin carotenoid levels for estimating the FV intake.

## 2. Methods

To identify studies examining factors that may influence skin carotenoids, a PubMed search of all English language articles for all years until November 2022 containing the following terms was performed: “humans [mh] AND (skin carotenoids OR dermal carotenoids) AND (fruits OR vegetables) NOT in-vitro”. Additional studies were identified by handsearching references included in the studies and review articles returned by this search. Studies were only included if they performed and reported the results of statistical tests to investigate potential differences in skin carotenoid levels or responses to dietary carotenoids associated with any of the covariates they assessed. Factors affecting blood carotenoids were identified using two recent reviews on the topic [28,29] and additional studies that addressed related content but were not included in these reviews [30,31,32,33,34,35,36,37,38,39,40,41,42].

## 3. Factors/Covariates Affecting Skin Carotenoid Levels

Many studies measuring skin carotenoid levels among children and adults using spectroscopy-based devices have investigated their potential relationships with a number of factors previously found to affect blood carotenoid concentrations, as well as others that are suspected to uniquely affect skin carotenoid levels. Shown in Table 1 and summarized in Figure 1 are the findings of these studies. The studies were stratified by the type of spectroscopic devices used where possible to account for potential disparities in results that may be present due to differences in methodology. The covariates considered included age, sex, body mass index (BMI), race/ethnicity, smoking status, alcohol consumption, carotenoid supplement intake, season, UV exposure, melanin/skin tone, medication use, dietary factors, and pregnancy or lactation. Following is a detailed discussion of the findings regarding each of these covariates.

Although there are a few studies that observed there was a positive correlation between age and skin carotenoid levels, most suggested they did not differ significantly by age [23,43,44,45,46,47,48,49,50,51,52,53,54,55,56,57,58,59,60,61,62,64,65,66,67]. Two studies on adults that controlled for dietary carotenoid/FV intake found that age did not significantly influence the skin carotenoid scores [55,64], while another found age was associated with significantly greater skin carotenoid levels [65]. The two studies on children which controlled for dietary intake had opposing results; age was significantly positively associated with skin carotenoid levels in one study [45] and negatively associated with skin carotenoid levels in the other [49]. Importantly, it should be noted that the former study was on a younger group of children aged 3–5, and the latter was on a group aged 2–12.

Overall, the included studies indicated that females frequently had significantly greater skin carotenoid values than males, although a few studies suggested the opposite [23,43,44,45,49,50,51,52,53,55,56,58,60,62,63,64,66,68,69,70,71,72,73,74]. Of the previous studies, only two on adults controlled for dietary carotenoid/FV intake. One on an older group of adults found that females had significantly higher skin carotenoid levels [55], while the other on a slightly younger group did not [64]. As for children, one study evaluated potential differences in skin carotenoid levels by sex after controlling for the intake of dietary carotenoid sources and found that there was no significant difference in the levels between sexes [45].

In the majority of studies on adults, participants with a greater BMI consistently had significantly lower skin carotenoid scores, but most studies on children did not report similar findings [23,43,44,45,49,50,51,55,56,59,61,62,64,65,68,69,72,73,74,75,76,77,78,79]. Four of the seven studies on adults which controlled for dietary carotenoid/FV intake also found that BMI was significantly inversely associated with skin carotenoid levels [55,61,65,78], while the others observed no significant association between the two [64,76,79]. Regarding children, of the three studies that controlled for dietary carotenoid/FV intake, one on younger (3–5 year) children found that BMI was significantly inversely associated with skin carotenoid levels [45], while another on a group with a broader age range (2–12 year) found no significant association of BMI with skin carotenoid levels [49], and a third found changes in skin carotenoid scores were not significantly influenced by changes in BMI percentiles following an intervention to promote FV intake in a small group of 4th grade students [75].

Most of the available studies did not observe significant differences in skin carotenoid values between races/ethnicities [23,44,45,49,50,52,55,60,61,63,64,65,68,69,71,72,76]. Two of the four studies on adults that controlled for dietary carotenoid/FV intake observed that Asian participants had significantly greater values than white or other non-white participants [65,76], while the others found no significant differences in skin carotenoid levels between races/ethnicities. Just two of the studies on children controlled for dietary carotenoid/FV intake, and their results revealed that in younger children (aged 3–5), those who were non-Hispanic White had significantly greater skin carotenoid scores than those who were Hispanic [45], while there were no significant differences in carotenoid levels between ethnicities among a group containing children from a larger age range (ages 2–12) [49].

One study carried out on low-income adults investigated potential differences in skin carotenoid levels according to the pregnancy and breastfeeding status of female participants [61]. This study observed that pregnant women did not have significantly different values compared to non-pregnant women, nor did breastfeeding women had significantly greater values than non-breastfeeding women. Notably, only the assessment of potential differences in skin carotenoid levels according to the breastfeeding status included an adjustment for the dietary intake of carotenoid-rich foods.

In the six studies that investigated the relationship between skin tone or melanin content and skin carotenoid levels, neither was significantly associated with skin carotenoid levels among adults, whether or not adjustments for dietary carotenoid/FV intake were included [23,58,64,67,76,79]. However, since skin carotenoid measurements were mainly conducted at the palm or fingertips in these studies, which are areas with little melanin, these findings may not apply to other areas of the skin.

Overall, most studies found that smokers have significantly lower skin carotenoid levels than non-smokers [23,55,56,58,60,61,62,65,66,76,80]. The results from studies that adjusted for dietary carotenoid/FV intake were largely similar. Four of these found that adults who currently or previously smoked had significantly lower skin carotenoid levels than non-smokers [55,65,78,80], while two others found that skin carotenoid levels did not differ significantly according to the participants’ smoking status [61,76].

One study that investigated potential differences in skin carotenoid values according to participants’ drinking status found that drinkers did not have significantly different skin carotenoid levels than non-drinkers [23]. However, this study did not analyze whether the lack of a significant difference was maintained after adjustment for the dietary intake of foods rich in carotenoids. In addition, another study on a small group of adult males found that skin carotenoid concentrations decreased significantly following the ingestion of alcohol [81].

Only a few studies have examined potential differences in skin carotenoids associated with medication use and have yielded conflicting findings [55,66]. One of these studies found that in older adults (aged 50–75), the use of any cholesterol, hypertension, or diabetes medication was associated with a significantly lower skin carotenoid score after adjustment for dietary carotenoid/FV intake [55]. The other, which was on Japanese adults aged 22–90 and did not include a similar adjustment, found that participants using lipid-lowering drugs had significantly greater skin carotenoid scores, which the authors noted may have been due to dietary modifications typically recommended upon the initiation of these medications [66]

At present, two studies have analyzed the relationship between recent/regular sunlight exposure and skin carotenoid levels, both of which were adjusted for dietary carotenoid/FV intake. These studies found that participants with recent or greater regular UV exposure had significantly lower skin carotenoid levels than those without recent/regular exposure [76,80]. Likewise, two studies have investigated possible differences in skin carotenoid levels by season and observed that the skin carotenoid levels of participants whose measurements were obtained in the fall and summer were significantly greater than those of participants whose measurements were recorded in the winter [51,76]. Furthermore, the skin carotenoid levels of individuals who had multiple measurements taken throughout the year were also significantly greater in the summer and fall compared to the winter [76].

Very few studies have examined the potential impact of dietary factors aside from fruit and vegetable intake on skin carotenoid levels. A study on older adults aged 50–75 found that only vitamin C intake was significantly positively associated with skin carotenoid values after adjustment for the intake of dietary sources of carotenoids [55]. Conversely, another study on children aged 2–12 observed that participants with a greater fat-to-energy intake ratio had significantly higher skin carotenoid levels [49]. As for supplements, two studies found individuals using lutein supplements had significantly greater skin carotenoid levels [58,62], and two others found that supplementation of beta-carotene elicited significant increases in skin carotenoid concentrations after accounting for the dietary FV/carotenoid intake [82,83].

## 4. Summary of Factors Affecting Skin Carotenoids and Comparison to Findings for Blood Carotenoids

Since it is already known that the dietary FV/carotenoid intakes influence the skin carotenoid levels, it is important to try to rule out the possibility that such factors are associated with them only because they are also associated with differences in FV/carotenoid intake. Furthermore, considering that before carotenoids can be deposited into skin tissue, they must be absorbed, metabolized, and transported in the blood, examining factors that are known or suspected to affect blood carotenoid concentrations may help identify those that may also be important to consider when conducting research seeking to validate skin carotenoid measurements for their ability to estimate the dietary FV intake. Therefore, found in Table 2 and discussed below is a summary of findings from studies examining factors affecting skin and blood carotenoids, focusing on those that included some form of control or adjustment for dietary FV/carotenoid intake where possible.

As expected, most of the factors previously found to affect blood carotenoid concentrations independent of the dietary carotenoid/FV intake were also found to associate with differences in skin carotenoids, although the amount of research pertaining to the latter was sparse. These factors included sex, BMI, smoking, carotenoid supplement intake, alcohol intake, and medication use. Specifically, males tended to have significantly lower blood or skin carotenoid levels [30,55], as did those with a greater BMI [30,45,55,61,65,78,84], smokers [38,55,65,78,80,112], and those taking certain medications [55,113,114], though some studies on adults and children failed to detect a significant association between BMI and skin carotenoid concentrations [49,64,74,75,79]. A few studies have shown supplemental carotenoids can increase the skin carotenoid levels [82,83]; however, it is not currently known whether the magnitude of these increases is greater than that produced by a similar intake from some food sources of carotenoids as research has shown is the case for blood carotenoid concentrations [122,123,124,125]. Regarding BMI, controlled feeding studies demonstrated that a greater BMI is associated with reduced blood carotenoid responses [30]. Given their lipophilic nature, an increased uptake of carotenoids by the adipose tissue is perhaps the most likely explanation for these findings [135]. However, one study observed that greater BMI and visceral fat were only associated with significantly lower blood carotenoid concentrations in adult women [136], and another on children found that a greater carotenoid intake via supplementation may contribute to reductions in BMI and body fat [137], which may explain some of the observed heterogeneity in findings. Accordingly, further research may be required to better characterize the influence of potential sex-specific differences on blood and skin carotenoids as well as the bidirectional nature of their relationship. Additionally, while specific medications such as statins and lipase inhibitors can reduce blood carotenoid concentrations [113,114,115], the only study that suggested medication use may decrease skin carotenoid levels did not focus on a specific medication, but rather on the use of any cholesterol, hypertension, or diabetes medication [55]. Therefore, additional studies designed to ascertain the influence of more specific types or classes of drugs on skin carotenoid levels are warranted.

In addition, some factors appear to have significant effects on blood carotenoid concentrations but either were not found to significantly affect or have yet to be studied with regard to their potential impacts on skin carotenoid levels. These included pregnancy/lactation status [61], genetics, menstrual cycle phase, health status, dietary fat and fiber intake [49,55], the food matrix, and plant stanol/sterol intake. A few studies have observed that women’s blood carotenoid concentrations are significantly greater in the third trimester of pregnancy [85,86] and lower shortly after initiating breastfeeding [87], but it has not been ruled out that these changes were due to dietary factors. In a study examining the potential influences of various factors on the skin carotenoid levels of low-income adults, pregnancy was not associated with a significant difference in the skin carotenoid levels, nor was breastfeeding after adjustment for FV/carotenoid intake [61]. Similarly, while a number of controlled feeding trials revealed that consumption of fat with carotenoid-rich foods may facilitate greater absorption and blood carotenoid responses [126,127,128], and a cross-sectional analysis found a very high fat intake was associated with significantly lower blood carotenoids at similar levels of FV intake [34], the only study that evaluated the association between fat intake (total, saturated, and unsaturated) and skin carotenoid levels found they were not significantly associated with each other after adjustment for dietary FV/carotenoid intake [55]. These findings suggest that the relationship between fat intake and carotenoid absorption may not be as straightforward as initially believed. It appears that when limited dietary carotenoids are consumed, a moderate increase in fat intake could promote greater absorption, but when large quantities are consumed, a greater fat intake may not further increase their absorption. As Marhuenda-Muñoz et al. (2021) hypothesized, this apparent duality could be related to the mechanism of carotenoid absorption. Though the co-consumption of moderate amounts of fat may enhance the bioavailability of carotenoids by increasing their extractability and micellarization, the intake of very large quantities of fat may hinder their bioavailability by causing the saturation of mixed micelles or competition for transporters. Nonetheless, further research is needed to better characterize the relationship between fat intake and skin carotenoid levels. Additionally, although carotenoids in “softer” processed foods have been found to be better absorbed and increase blood carotenoid concentrations to a greater extent than those in less processed foods, whether this translates to greater differences in skin carotenoids has not been determined. Finally, as previously discussed, a number of single-nucleotide polymorphisms, the phase of a woman’s menstrual cycle, the presence of certain diseases, and plant stanol/sterol intake can influence the blood carotenoid concentrations, but the impacts of these factors on skin carotenoid levels have yet to be investigated [35,88,89,90,91,92,93,94,95,96,97,98,99,100,101,102,103,104,105,106,107,108,109,110,111,117,118,119,120,121,134].

A few other factors that may influence the skin carotenoid levels independently of dietary FV/carotenoid intake according to existing research are race/ethnicity, season, vitamin C intake, and recent sun exposure. In a study on adults aged 21–65, recent sun exposure was associated with significantly lower skin carotenoid levels even though the participants’ levels were significantly lower in the winter compared to other seasons [76]. Likewise, another large study observed that greater usual sunlight exposure was associated with significantly lower skin carotenoid levels [80]. Therefore, despite findings from a previous study that skin yellowness (used as a proxy for skin carotenoid levels) increased in both sun-exposed and -unexposed skin areas following an increase in dietary carotenoid intake [138], these results indicate recent sun exposure may decrease the degree to which skin carotenoid levels correlate with FV intake. This finding is likely due to the fact carotenoids possess antioxidant properties and react with free radicals generated following exposure to UV light [139,140]. Regarding the finding that skin carotenoid levels are lower in the winter even after adjustment for dietary FV intake, this may be explained by the existence of differences in specific FV choices that could affect the carotenoid levels rather than the absolute intake, which could be explored in future studies. In addition, adult Asians had significantly greater skin carotenoid levels than adults of other races/ethnicities in some studies [65,76], but a few others found there were no significant differences in carotenoid levels across races/ethnicities [55,64]. Additionally, one study on children aged 3–5 found that non-Hispanic white participants had significantly greater skin carotenoid levels than Hispanic participants [45], but another on children aged 2–12 found no significant differences in carotenoid levels by ethnicity [49]. Finally, the finding that vitamin C intake may be associated with increased skin carotenoid concentrations independently of the dietary carotenoid intake is not unexpected. Previous research has established that vitamin C, but not vitamin E, can convert water-insoluble carotenoid radical cations generated via reactions between carotenoids and free radicals back to the parent carotenoids [141] and thereby may decrease the rate of their degradation in the skin. Nonetheless, the small number of studies concerning each of these factors highlights the exigency of further research into their effects on skin carotenoid levels.

Lastly, age and melanin/skin tone have not been found to significantly affect the skin carotenoid levels after adjustment for dietary FV/carotenoid intake in most studies conducted thus far [49,55,64], though one study on adults [65] and one on young children aged 3–5 [45] observed a significant positive association between age and skin carotenoid levels. Although there has been concern regarding the potential for melanin/skin tone to confound the optical assessments of skin carotenoids, these results, in conjunction with findings from previous research demonstrating that the skin carotenoid levels in both lightly and highly pigmented areas of the skin with low sun exposure increase significantly following carotenoid supplementation [142], indicate it may not pose as much of an issue as initially expected. However, it should be noted that the skin carotenoid measurements in the studies discussed were mainly performed on the palm or fingertips, which are areas with little melanin; so, these findings may not apply to other areas. Additionally, skin melanin/tone and age also require further attention in future research, given the existing studies are limited in number and encompassed a narrow breadth of ages and skin tones.

In addition, it is important to acknowledge that the type of spectroscopy device used and the methods for performing skin carotenoid measurements were markedly heterogenous across the included studies and may have influenced the nature of the observed associations between the abovementioned factors and skin carotenoids. Regarding the type of device used, though it was difficult to make comparisons due to differences in the sizes and racial/ethnic composition of the groups studied, we did not observe any clear inconsistencies between findings for any of the factors investigated where studies using both devices were available. Regardless, if possible, it would be desirable for future studies to use both devices and compare the results concerning the associations between various factors and skin carotenoid levels so as to identify any potential differences between them.

Furthermore, alongside the factors investigated in the studies included in this review, we feel there are some which may be worth considering in future research. Previous work has identified that carotenes may be favored over xanthophylls with regard to deposition in the skin [143], and therefore research directly investigating the influence of specific carotenes and xanthophylls from food or supplements on total skin carotenoid levels may help uncover the relative contribution of each and best identify the dietary constituents most likely to be reflected by skin carotenoid levels. In addition, due to the impact of physical exercise on metabolism, serum lipid concentrations, and overall antioxidant capacity, it would be interesting to see future studies on the impact of physical exercise type and frequency on skin carotenoid concentrations. Finally, further research into the effects of certain disease states and medications not previously studied on skin carotenoid levels may also turn out to be fruitful.

## 5. Conclusions

Due to the convenience, cost-effectiveness, and timeliness of measuring skin carotenoid levels with spectroscopy-based devices, there has been a growing interest in using them to estimate the FV intake. At present, many studies have found that skin carotenoid levels measured with these devices are significantly positively associated with self-reported FV intake and blood carotenoid concentrations [26,144]. However, there are several factors that may affect the skin carotenoid levels independently of the dietary intake of carotenoids which must be identified and considered in order to optimize their ability to estimate the FV intake among diverse populations. Accordingly, by surveying the available literature, this review summarized findings from studies on blood and skin carotenoids to determine factors that may affect the skin carotenoid levels, highlight gaps in knowledge, and offer guidance for future researchers seeking to validate their ability to estimate the FV intake.

In the studies reviewed, sex, BMI, race/ethnicity, smoking status, carotenoid supplement intake, and recent sun exposure were most consistently associated with significant differences in skin carotenoids after adjustments for dietary carotenoid/FV intake and should therefore represent the minimally sufficient set of covariates to be considered when using spectroscopy-measured skin carotenoid levels to estimate the dietary FV intake. Additionally, some evidence suggests that alcohol intake, medication use, and season may affect skin carotenoids independently of the dietary FV/carotenoid intake, but more research is required to confirm these findings. Furthermore, there are some factors, such as genetics, menstrual cycle stage, and health status, whose independent impacts on skin carotenoids have not been studied at all, despite previous findings demonstrating that they affect blood carotenoids. Most importantly, of the studies seeking to validate spectroscopy-based devices for the estimation of FV intake, very few included any form of control for the abovementioned factors.

In closing, while the measurement of skin carotenoid levels using spectroscopy-based devices appears to be a promising method for assessing the efficacy of nutrition-based policies, programs, and interventions to promote FV intake, there is a need for well-controlled clinical studies to both characterize additional factors which may influence them independently of the dietary FV intake and validate this method in groups with diverse racial/ethnic backgrounds and physiological conditions, while including considerations for said factors.

## Figures and Tables

**Figure 1 nutrients-15-02156-f001:**
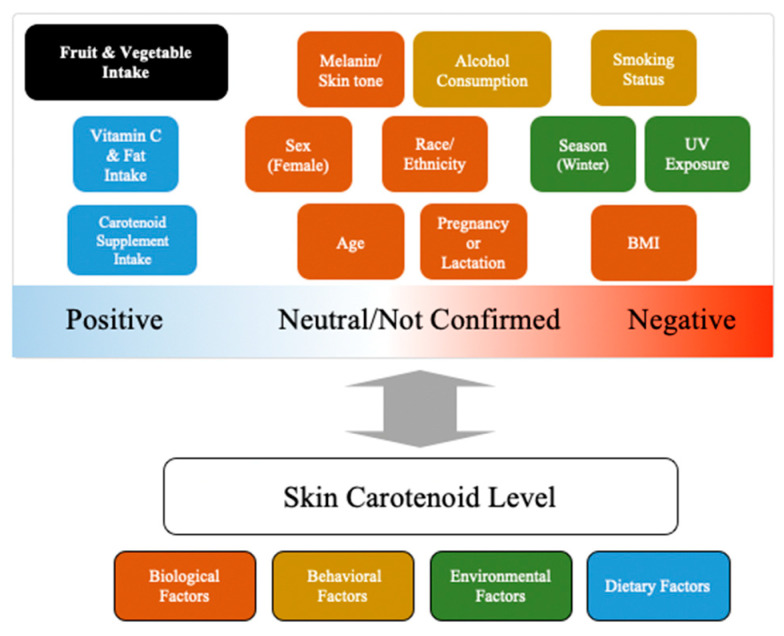
Factors that influence skin carotenoid levels in humans. (Note: the influence of FV intake was not considered in this review due to the fact that it is well accepted there is a moderate correlation between FV intake and skin carotenoid levels).

**Table 1 nutrients-15-02156-t001:** Summary of factors that influence skin carotenoid levels.

	Findings
Age	*Children* RRS [43] Age NS in 9–12 year US children (n = 128) [44] Age NS in US children/adolescents 5–17 year (n = 45) [45] Age + in US children 3–5 year in low-income homes (n = 381) * [46] Age + US infants/children in <1–7 year (n = 51) [47] Gestational Age + in US preterm infants (n = 16) [48] Age − in US formula-fed and NS in breast-fed male infants, birth–9 w (n = 40) [49] Age − in US children 2–12 year in low-income homes (n = 177) * [50] Age NS in healthy US children 3–5 year (n = 209) RS [51] Age NS in US Latino children 10–14 year (n = 195) [52] Age NS in US children 9–12 year (n = 143) [53] Age + in US children 3–5 year in low-income homes (n = 112) *Adults* RRS [23] Age NS in US adults 21–65 year (n = 74) [54] Age NS in US adults 50–85 year with AMD (n = 44) [55] Age NS in Singaporean adults 50–75 year (n = 103) * [56] Age NS in German adults 18+ year (n = 151) [57] Age NS in German adults 19–79 year (RRS and RS, n = 33) RS [58] Age NS in Japanese eye clinic patients 6–98 year (n = 569) [59] Age NS in older US adults 65–86 year (n = 95) [60] Age NS in US adults 20–84 year (n = 80) [61] Age NS in low-income US adults (n = 287) [62] Age NS in Japanese adults 16–97 year (n = 985) [63] Age NS in US adults 18+ year (n = 136) [64] Age NS in healthy US adults 18–65 year (n = 213) * [65] Age + in a diverse sample of NZ adults 16+ year (n = 571) * [66] Age + in Japanese adults 22–90 year (n = 1812) [67] Age + in healthy Koreans and Germans 7–75 year (n = 714)
Sex	*Children* RRS [44] Sex NS in US children/adolescents 5–17 year (n = 45) [45] Sex NS in US children 3–5 year in low-income homes (n = 381) * [49] Sex NS in US children 2–12 year in low-income homes (n = 177) [68] Sex NS in US children 5–17 year (n = 58) [50] Sex NS in healthy US children 3–5 year (n = 209) [43] Males + in US children 9–12 year (n = 128) RS [52] Sex NS in US children 9–12 year (n = 143) [51] Sex NS in US Latino children 10–14 year (n = 195) [69] Males + in US pre-school (n = 112) and middle-school (n = 94) children; Sex NS in US high school children (n = 58) [53] Males + in US children 3–5 year in low-income homes (n = 112) [70] Sex NS in Japanese children 10 year (n = 315) *Adults* RRS [23] Sex NS in US adults 21–65 year = (n = 74) [55] Females + in Singaporean adults 50–75 year (n = 103) * [56] Females + in German adults 18+ year (n = 151) RS [71] Sex NS in low-income older US adults > 60 year (n = 154) [72] Sex NS in US college students 18–25 year (n = 66) [73] Sex NS in US college students (n = 40) [64] Sex NS in healthy US adults 18–65 year (n = 213) * [66] Females + in Japanese adults 22–90 year (n = 1812) [62] Females + in Japanese adults 16–97 year (n = 985) [58] Females + in Japanese eye clinic patients 6–98 year (n = 569) [60] Males + US adults 20–84 year (n = 80) [63] Males + in US adults 18+ year (n = 136) [74] Females + in healthy Japanese adults 20+ year (n = 811)
BMI	*Children * RRS [44] BMI% NS in US children/adolescents 5–17 year (n = 45) [45] BMI − in US children 3–5 year in low-income homes (n = 381) * [49] BMI NS in US children 2–12 year in low-income homes (n = 177) * [68] BMI NS in US children 5–17 year (n = 58) [43] BMI − (Normal < Overweight < Obese) in US children 9–12 year (n = 128) [50] BMI NS in healthy US children 3–5 year (n = 209) [75] BMI NS in US 4th grade students (n = 30) * RS [51] BMI NS in US Latino children 10–14 year (n = 195) [69] BMI NS in US preschool (n = 112), middle school (n = 94), and high school children (n = 58) *Adults * RRS [23] BMI NS in US adults 21–65 year (n = 74) [76] BMI NS in US adults 21–65 year (n = 74) * [56] BMI NS (*p* < 0.1) in German adults 18+ year (n = 151) [77] BMI NS in Thai adult health professionals (n = 29) [55] BMI − in Singaporean adults 50–75 year (n = 103) * [78] BMI − in US women with breast cancer 18–90 year (n = 102) * RS [72] BMI NS in US college students 18–25 year (n = 66) [64] BMI NS in healthy US adults 18–65 year (n = 213) * [59] BMI − in older US adults 65–86 year (n = 95) [61] BMI − in low-income US adults (n = 287) * [73] BMI − in US college students (n = 40) [65] BMI − in a diverse sample of NZ adults 16+ year (n = 571) * [62] BMI − in Japanese adults 16–97 year (n = 985) [79] BMI NS in US African American college students 18–30 year (n = 98) * [74] BMI − in healthy Japanese adults 20+ year (n = 811)
Race/Ethnicity	*Children* RRS [44] Ethnicity NS in US children/adults 5–17 year (n = 45) [45] White non-Hispanic > Hispanic in US children 3–5 year in low-income homes (n = 381) * [49] Ethnicity NS in US children 2–12 year in low-income homes (n = 177) * [68] Ethnicity NS in US children 5–17 year (n = 58) [50] Ethnicity NS in healthy US children 3–5 year (n = 209) RS [69] Ethnicity NS in US preschool (n = 112), middle school (n = 94), and high school children (n = 58) [52] Ethnicity NS in US children 9–12 year (n = 143) *Adults* RRS [23] Ethnicity NS in US adults 21–65 year (n = 74) [55] Ethnicity NS in Singaporean adults 50–75 year (n = 103) * [76] Asian > White in US adults 21–65 year (n = 74) * RS [60] Ethnicity NS in US adults 20–84 year (n = 80) [61] Ethnicity NS in low-income US adults (n = 287) [72] Ethnicity NS in US college students 18–25 year (n = 66) [63] Ethnicity NS in US adults 18+ year (n = 136) [64] Ethnicity NS in healthy US adults 18–65 year (n = 213) * [65] Asian > other ethnicities in a diverse sample of NZ adults 16+ year (n = 571) * [71] Non-white > white in low-income older US adults > 60 year (n = 154)
Pregnancy/Lactation	RS [61] Pregnancy NS and breastfeeding NS in low-income US adults (n = 287) *
Melanin/Skin Tone	RRS [58] Skin melanin NS in an ethnically diverse sample of US adults (RRS and RS, n = 160) [23] Darker skin tone NS in US adults 21–65 year (n = 74) [76] Skin tone NS in US adults 21–65 year (n = 74) * [57] Skin tone NS in German adults 19–79 year (RRS & RS, n = 33) RS [79] Skin tone NS in US African American college students 18–30 year (n = 98) * [64] Skin melanin NS in healthy US adults 18–65 year (n = 213)
Smoking Status	RRS [23] Smoking NS in US adults 21–65 year (n = 74) [76] Smoking NS in US adults 21–65 year (n = 74) * [55] Smoking − in Singaporean adults 50–75 year (n = 103) * [56] Smoking − in German adults 18+ year (n = 108) [80] Smoking − in healthy British subjects (n = 1375) * [78] Smoking − in US women with breast cancer 18–90 year (n = 102) * RS [60] Smoking NS US adults 20–84 year (n = 76) [58] Smoking − in Japanese eye clinic patients 6–98 year (n = 569) [61] Smoking NS in low-income US adults (n = 287) * [65] Smoking − in a diverse sample NZ adults 16+ year, NS in multiple regression (n = 571) * [66] Smoking − in Japanese adults 22–90 year (n = 1812) [62] Current < Past < Never smokers in Japanese adults 16–97 year (n = 985)
Alcohol Consumption	RRS [23] Alcohol drinker NS in US adults 21–65 year (n = 74) [81] Alcohol consumption − in German adult males 21–54 year (n = 6) *
Medications	RRS [55] Cholesterol, hypertension, and diabetes medications − in Singaporean adults 50–75 year (n = 103) * RS [66] Hypolipidemic agents + in Japanese adults 22–90 year (n = 1812)
Season	RRS [76] Spring/Autumn and Summer > Winter in US adults 21–65 year (n = 74) * RS [51] Fall > Winter in US Latino children 10–14 year (n = 195) *
UV Exposure	RRS [76] Recent sun exposure – in US adults 21–65 year (n = 74) * [80] High sunlight exposure – in healthy British subjects (n = 1375) *
Carotenoid Supplement Intake	RRS [82] Beta-carotene + lycopene supplement + in healthy German female adults 21–72 year (n = 129) * RS [58] Lutein supplements + in Japanese eye clinic patients 6–98 year (n = 569) [62] Lutein supplements + in Japanese adults 16–97 year (n = 985) [83] Beta-carotene supplement + in healthy German female adults 20–45 year (n = 12) *
Nutrient Intake	RRS [55] Vitamin C intake + in Singaporean adults 50–75 year (n = 103) * [49] Fat intake (% kcal) + in US children 2–12 year in low-income homes (n = 177) *

+: Significantly positively associated with skin carotenoids. −: Significantly negatively associated with skin carotenoids. NS: Non-significantly associated with skin carotenoids. *: association with specified factor adjusted/controlled for dietary carotenoids/FV.

**Table 2 nutrients-15-02156-t002:** Summary of factors or covariates that influence blood/skin carotenoids in adults and children.

Factor/Covariate	Blood Carotenoids	Skin Carotenoids	Summary
Biological Factors			
Age	NS: [30]	Children +: [45] NS: [49] Adults NS: [55,64] +: [65]	In most of the available studies on adults, age does not appear to significantly impact blood or skin carotenoid levels. One study on a group of children aged 3–5 found that age was significantly positively associated with skin carotenoids, but another on a group aged 2–12 did not.
Sex	Females +: [30]	Children NS: [45] Adults Females +: [55] NS: [64]	Studies have consistently observed that females have significantly greater blood carotenoid concentrations independently of dietary carotenoid intake, but conflicting results exist with respect to the potential differences in skin carotenoid levels between males and females.
BMI	−: [30,84]	Children −: [45] NS: [49,75] Adults −: [55,61,65,78] NS: [64,76,79]	Most studies have found that a greater BMI is associated with significantly lower blood carotenoids, but the relationship between BMI and skin carotenoids is not quite as consistent, especially among children. This may be at least in part due to the heterogeneity of the size and composition of the studied populations.
Race/Ethnicity	NS: [55]	Children NH White > Hispanic: [45] NS: [49] Adults Asian +: [65,76] NS: [55,64]	Most of the existing research has not identified significant differences in skin carotenoids across racial/ethnic groups after accounting for dietary carotenoid and FV intake, though a few studies have observed that among adults, Asian individuals have significantly greater skin carotenoids than other racial/ethnic groups.
Pregnancy or Lactation	Pregnancy (third trimester) +: [85,86] Lactation –: [87]	Breastfeeding NS: [61] Pregnancy NS: [61]	Blood carotenoids have been found to increase significantly in the third trimester of pregnancy and decrease during lactation, though the potential of these observations being a result of dietary changes was not ruled out. The limited data on skin carotenoids suggest breastfeeding may not affect skin carotenoids independently of FV/carotenoid intake, and pregnancy is not associated with a significant difference in skin carotenoid levels (without dietary adjustment).
Genetics	Many gene variants influence blood carotenoids or modify blood responses to carotenoid ingestion: [88,89,90,91,92,93,94,95,96,97,98,99,100,101,102,103,104,105,106,107,108]	Not yet studied	Quite a few gene variants have the ability to affect blood carotenoid concentrations and responses to dietary carotenoid intake, but the implications for skin carotenoids have not been studied directly.
Menstrual Cycle	Lowest during menses, highest in the midluteal phase: [109,110,111]	Not yet studied	Studies have revealed that blood carotenoids decrease during menses and are greatest in the midluteal phase. No research has been conducted to assess how skin carotenoids may vary across stages of the menstrual cycle.
Melanin/Skin Tone	Not yet studied	NS: [64,76,79]	A few studies have found that melanin or skin tone does not significantly affect skin carotenoids when adjusting for dietary carotenoid or FV intake, but more research on larger and more diverse samples is warranted.
Behavioral Factors			
Smoking Status	−: [38,112]	−: [55,65,78,80] NS: [61,76]	Current and previous smokers tend to have significantly lower blood and skin carotenoid levels.
Alcohol Consumption	−: [40]	−: [81]	One interventional study found that increasing alcohol intake elicits a significant decrease in blood carotenoids, and a very small pilot study found that alcohol consumption significantly decreased skin carotenoid levels. Future research should be conducted to confirm these findings.
Medication Use	Statins -: [113,114] Lipase Inhibitor -: [115]	Cholesterol, hypertension, and diabetes medications −: [55]	Statins and lipase inhibitors significantly decrease blood carotenoids, and one study on skin carotenoids found that medication use (including statins) is associated with lower levels after adjustment for dietary carotenoid intake. Additional research is needed to better quantify the influence of medications on skin carotenoids.
Health Status	CVD −: [116,117] Diabetes/IR −: [118,119,120] Chronic Cholestasis −: [121]	Not yet studied	Many studies have found CVD, diabetes/IR, and chronic cholestasis are associated with lower blood carotenoid levels; however, dietary carotenoid intake was not accounted for. Currently, there is no research on the effect of these or other diseases on skin carotenoid levels.
Environmental Factors			
Season	Not yet studied	Winter −: [76]	In one study, skin carotenoid levels were significantly higher in spring/summer and autumn than in winter after accounting for dietary carotenoid intake. The potential seasonal influence on skin carotenoids should continue to be investigated to further confirm these findings.
UV/Sun Exposure	Not yet studied	Recent sun exposure −: [76,80]	Two studies found that participants with recent or greater usual sun exposure had significantly lower skin carotenoid levels. These findings should be confirmed in future research.
Dietary Factors			
Carotenoid Supplement Intake	Supplement carotenoid bioavailability is frequently greater than food bioavailability [122,123,124,125]	Carotenoid supplement intake +: [82,83]	Interventional studies have shown carotenoid supplement bioavailability is higher than that of carotenoids from certain foods. While a few studies have suggested carotenoid supplement ingestion significantly increases skin carotenoid levels, it remains unclear if these increases are greater than those elicited by ingestion of similar amounts of carotenoids from food.
Dietary Fat Intake	Acute feeding studies suggest co-consumption of fat increases carotenoid absorption (UFAs > SFAs), but associations between self-reported intakes of fat and blood carotenoids have been inconsistent [34,55,126,127,128]	Intake of total, saturated, and unsaturated fats (g/d) NS: [55]	Acute feeding studies found that co-consumption of fat can increase carotenoid bioavailability. However, one study found that greater habitual fat intake does not significantly influence blood or skin carotenoids independently of dietary carotenoid intake, and another found that a very high fat intake is associated with reduced blood carotenoid concentrations compared to a low/moderate fat intake. Additional research should seek to clarify whether or not fat intake meaningfully affects skin carotenoids.
Dietary Fiber Intake	One study found that fiber enrichment of meals may decrease carotenoid bioavailability, but another found a greater habitual dietary fiber intake did not significantly affect serum carotenoid concentrations [55,129]	Fiber NS: [55]	One interventional study found that the addition of fiber to meals may decrease carotenoid bioavailability, though in other research a greater dietary fiber intake did not significantly affect blood or skin carotenoids. Upcoming research should seek to confirm these findings.
Food Matrix	Carotenes from “softer” processed foods are more bioavailable than those from minimally processed foods [122,130,131,132,133]	Not yet studied	Processing of carotenoid-rich foods into “softer” forms seems to increase their bioavailability and thus their ability to increase blood carotenoid levels, but whether this also translates to skin carotenoid levels has yet to be studied.
Plant Stanols/Sterols	Intake of plant stanols/sterols appears to decrease absorption and plasma concentrations of blood carotenoids [35,134]	Not yet studied	Multiple interventional trials have found that plant stanols/sterols can decrease blood carotenoid concentrations, but no studies have investigated their influence on skin carotenoid levels.

+: Significantly positively associated with metric in column, −: Significantly negatively associated with metric in column, NS: Non-significantly associated with metric in column. FV: Fruit and vegetables, NH: Non-Hispanic, CVD: Cardiovascular Disease, IR: Insulin Resistance.

## Data Availability

No new data were created or analyzed in this study. Data sharing is not applicable to this article.

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
