# Peer review of "A Narrative Review of Factors Associated with Skin Carotenoid Levels"

_nutrients, 2023, doi:10.3390/nu15092156_

Round 1

Reviewer 1 Report (New Reviewer)

In this review, Madore and colleagues evaluated the potential of skin carotenoid concentrations as a biomarker for fruits and vegetables intake and reviewed the role of confounding factors that may affect the rigor of effective estimates. Notably, the intake of nutrients such as vitamin C, dietary fat and carotenoid supplements were found to display positive associations while smoking status, UV exposure, BMI and measurements during winter were negatively associated with skin carotenoid levels. Overall, the authors performed a comprehensive review of the available evidence and the manuscript was well-written and organized. Please find my specific questions, comments, and suggestions listed below:

1) As described by the authors, there are two main spectroscopic methods used to estimate skin carotenoids, RS and RRS. I would recommend a stratification of the studies based on the mode of assessment since inherent differences in methodology may influence the type of confounding factors identified.

2) In a similar context, since each of these studies likely used different devices for analysis, the rigor of measurements, consistency of instrument, etc. may also influence the accuracy of estimates and covariates identified. This in my opinion should be addressed as well.

3) One of the advantages of using skin carotenoids is to provide estimates of fruits and vegetables intake without the need for subjective survey-based methods. However, as identified by the authors, it seems like nutritional factors such as vitamin C and fat intake may modulate skin carotenoid levels. Hence, it makes me question the utility of skin carotenoid levels as biomarkers since these nutritional factors will need to be acquired from dietary data to begin with.

4) A factor I feel the authors overlooked is the type of carotenoids and its association with skin carotenoids. For example, the more hydrophobic carotenes may be favored over xanthophylls for storage in the skin, suggesting that the choice of fruit and vegetables may also matter.

5) Likewise, it should also be noted that there are fruits and vegetables with little to no carotenoids present. However, since antioxidants like vitamin C may influence skin carotenoid levels, would other dietary antioxidants such as tocopherols and polyphenols also affect circulating and skin carotenoid levels? 

Author Response

Thank you for your time and feedback, please see attached. 

Reviewer 2 Report (New Reviewer)

This manuscript is a thorough and interesting review article which will be useful to researchers interested in carotenoids and their dietary uptake into the skin. I have only a few minor comments/suggestions to help improve the manuscript and, therefore, recommend that it is accepted for publication after only minor changes.

My first comment relates to lines 204-207; The fact that vitamin C intake has been shown to positively influence carotenoid skin levels, is perhaps not surprising given that carotenoid radical cations have been shown to be repaired by vitamin C, regenerating the parent carotenoid. These carotenoid radicals are likely to be formed from reactive oxygen species generated in the skin, possibly photochemically from sunlight exposure. Therefore, it is probably worth referencing some of the articles that show this protective effect of vitamin C. See, for example, work by George Truscott and also by Homer Black.

Similarly this work, and studies by others on carotenoids reacting with various radicals and reactive oxygen species, could also help to explain why sunlight exposure has a negative effect on skin carotenoid levels and the section on page 9 discussing sun exposure could be expanded.

I think expanding section 4, especially on the contradictions regarding fat consumption, % body fat and BMI could be a useful addition to the manuscript. As examples, it is known that carotenoid absorption is increased when fruit and vegetables are eaten with dietary fats, see, for example, Gärtner, Stahl and Sies, Lycopene is more bioavailable from tomato paste than from fresh tomatoes. Amer. J. Clin. Nutr. 66 (1997) 116-122, and ref 52 in this manuscript. Since this is the case and women generally have a higher % body fat than men, then it seems likely that they would have higher skin carotenoid levels. However, BMI seems to be negatively associated with skin carotenoids. Do the authors think that it is simply that those with higher BMIs are consuming less FV or could it be that more of the carotenoids are excreted without being absorbed if there is too high fat consumption in the diet?

Finally, lines 328-329 in the conclusion mention that certain factors have not been studied at all. Table 2 shows where there are factors that have only been studied for blood carotenoids and not skin, or vice versa, but are there more factors, which have not been studied for either, that the authors would recommend for future studies?

Author Response

Thank you for your time and feedback, please see attached. 

This manuscript is a resubmission of an earlier submission. The following is a list of the peer review reports and author responses from that submission.

Round 1

Reviewer 2 Report

This is a very comprehensive paper detailing important covariates of the plasma and skin carotenoid response to dietary intake. My comments are as follows:

1.       The title seems inappropriate for the content of the paper. The paper seems to be about important covariates of the plasma and skin carotenoid response to dietary carotenoids and/or the important covariates that those who are using plasma and skin carotenoids to approximate fruit and vegetable intake should take into account. So that should be a little clearer from the title.

2.       In the introduction, around lines 76-77, should a brief description of resonance Raman spectroscopy versus pressure-mediated reflection spectroscopy be provided?

3.       The stated purpose is: “Therefore, this review seeks to summarize the available research on factors that may affect skin carotenoid levels, discuss current uncertainties and gaps in knowledge, and provide guidance for future research efforts seeking to validate spectroscopy-measured skin carotenoid levels as a means of accurately estimating FV intake among various  populations.” However, a large proportion of the paper is about factors related to plasma carotenoids, so should the purpose statement change, both in the abstract and introduction?

4.       The purpose also states that this is a “review” but there are no methods detailing how the review was conducted. There is no rationale for why the particular covariates were selected, no databases or search terms listed, no PRISMA diagram, no description of double-coders or extraction forms, etc. The methods for the review seem to be absent.

5.       For several of the factors that affect plasma carotenoids, the same list is not examined for skin carotenoids. What is the rationale?  It seems like the exact same list of covariates should be examined for both plasma and skin carotenoids.

Reviewer 3 Report

Nutrients (ISSN 2072-6643)

Manuscript ID - nutrients-2244757

Type- Review

Title- Skin carotenoid levels as an Estimator of Fruit and Vegetable Intake: Current Knowledge and Future Directions

Authors- Matthew Phillip Madore , Jeong-Eun Hwang , Jin-Young Park , Seoeun Ahn , Hyojee Joung , Ock K. Chun *

Section - Nutrition Methodology & Assessment

Special Issue - Assessment of Nutritional Status and Nutrition in Chronic Diseases–Prevention, Treatment and Life Quality Management

The authors briefly summarize the literature related to dietary and non-dietary factors that influence skin carotenoid levels and discuss existing uncertainties and provide direction for future research aimed at further validating RRS and RS-measured skin carotenoid levels for estimating FV intake. The review is well written with the inclusion of upto date literature review.

1.       Fruits and vegetables are not the only sources of carotenoids how measuring skin carotenoids levels can estimate dietary FV intake

2.       The author may consider leaving out section 2.0-2.18 as it is well-known and many reviews discusses it and it dilutes the main content and more emphasis to be given in detail for section 3.0-3.11.

3.       The summary of findings and future directions are to be presented under separate subsections.

Round 2

Reviewer 1 Report

The authors have done an excellent job responding to my previous comments. My only remaining suggestions is that they update the title of Table 2 to be more accurate and specific. I suggest something like… “Summary of factors or covariates that influence blood/skin carotenoids in adults and children, after controlling for fruit and vegetable intake.”

Author Response

Reviewer 1, 
Thank you, we have very much appreciated your feedback. We have changed the title of Table 2 to better reflect its contents, which can be seen in the latest version of the manuscript.  

Reviewer 2 Report

The authors have done a lot to address the prior comments. However, there are lingering issues that remain.

 1.       The title is improved, but I suggest shortening to “A narrative review of factors associated with blood and skin carotenoid levels.”

2.       The paragraph added in the first part of the methods is informally written. Please use more formal writing. I suggest deleting the first sentence and starting with “This narrative review was guided by two recent reviews on the topic of factors associated with plasma carotenoid levels (cite Moran and Borel) and supplemented by additional research.”  Then expand on what the authors mean by “supplemented by additional research”…Is this the PubMed search?

3.       Did the authors then look at each of the factors that was associated with plasma carotenoids in these two reviews and look in the literature (PubMed) to see if there was evidence of these factors also influencing skin carotenoids? If so, please state this. If not, please state the reasons certain factors in the Moran and Borel review papers were examined for effects on skin carotenoids whereas others were not.

4.       The sentence “However, other research was also surveyed” (line 113) leaves the reader asking the question: “What other research?” Why was this other research surveyed? The way the methods are written, it seems like an unsystematic search was conducted, and then the authors just selected papers they liked to represent the entire body of literature on this topic. More work is needed in the methods section to show readers that a thorough, non-biased search was conducted, and that the factors examined in this review paper were selected based on a non-biased assessment of the literature. As it is currently written, I am not convinced. Please look at other narrative review papers and add more structure and clarity to the methods.

5.       The note in figure 1 “FV intake is already known for its strong correlation with blood carotenoids…” is not accurate. In most studies, self-reported FV intake is only modestly related to plasma (blood) carotenoids.

6.       In several instances, it would be helpful to cite appropriately after a particular phrase, versus at the end of the entire sentence. This would help future readers when trying to find particular studies. Below are just examples, there are many instances in the paper where this would be helpful.

a.       Lines 107 – 109: “Therefore, instead of performing an extensive literature search, the discussion was guided by two recent reviews on the topic (Add Moran and Borel references here) and supplemented by additional research. (Add references for additional research here).”

b.       Lines 444-446: “Importantly, it should be noted that the former study was on a younger group of children aged 3-5 (ADD REFERENCE HERE) and the latter was on a group aged 2-12, (ADD REFERENCE HERE) which may offer an explanation for this apparent inconsistency.”

c.       Lines 463 – 469: “Regarding children, of the three studies that controlled for dietary carotenoid/FV intake, one on younger (3-5 y) children found that BMI was significantly inversely associated with skin carotenoid levels, (ADD REFERENCE HERE) while another on a group with a  broader age range (2-12 y) found no significant association of BMI with skin carotenoid  levels,(ADD REFERENCE HERE) and a third found changes in skin carotenoid scores were not significantly influenced by changes in BMI percentiles following an intervention to promote FV intake in a small group of 4th grade students [168,173,197](ADD FINAL REFERENCE HERE).”

d.       Lines 485-487: “Three found that adults who currently or previously smoked had significantly lower skin carotenoid levels than non-smokers,(ADD REFERENCES HERE) while two others found that skin carotenoid levels did not differ significantly according to participants’ smoking status [178,182,198,201,202] (ADD REFERENCES HERE).

e.       Lines 537 – 540: “As for supplements, two studies found individuals using lutein supplements had significantly greater skin carotenoid levels,(ADD REFRENCES HERE) and two others found that supplementation of beta-carotene elicited significant increases in skin carotenoid concentrations [175,183,204,205](ADD REFERENCES HERE).

f.        Lines 584-588: “Specifically, males tended to have significantly lower blood and skin carotenoid levels,(ADD REFERENCES HERE) as did those with a greater BMI, (ADD REFERENCES HERE) smokers,(REFS) and those taking certain medications,(REFS) though some studies on adults and children failed to detect a significant association between BMI and skin carotenoid concentrations [168,173,178,182,185,186,197,198,200,201] (ADD REFERENCES HERE).

g.       Lines 648 – 650: “Additionally, one study on children aged 3-5 found that non-Hispanic white participants had significantly greater skin carotenoid levels than Hispanic participants,(ADD REFERENCE HERE) but another on children aged 2- 12 found no significant differences in levels by ethnicity [168,173] (ADD REFERENCE HERE).

h.       Lines 653 – 656: “Lastly, age (Add references here) and melanin/skin tone (Add references here) have not been found to significantly affect skin carotenoid levels in most studies conducted thus far, though one study on young children observed a significant positive association between age and skin carotenoid levels (Add references here)  [168,173,178,185,198,200].
